# Time to Play in Javanese Preschool Children—An Examination of Screen Time and Playtime before and during the COVID-19 Pandemic

**DOI:** 10.3390/ijerph20031659

**Published:** 2023-01-17

**Authors:** Soni Nopembri, Rizki Mulyawan, Puji Yanti Fauziah, Erma Kusumawardani, Indri Hapsari Susilowati, Lukman Fauzi, Widya Hary Cahyati, Tandiyo Rahayu, Terence Buan Kiong Chua, Michael Yong Hwa Chia

**Affiliations:** 1Department of Physical Education, Faculty of Sports and Health Sciences, Universitas Negeri Yogyakarta, Sleman 55281, Indonesia; 2Department of Sports Science, Faculty of Sports and Health Sciences, Universitas Negeri Yogyakarta, Sleman 55281, Indonesia; 3Nonformal Education Department, Faculty of Education Science, Universitas Negeri Yogyakarta, Sleman 55281, Indonesia; 4Department of Occupational Health and Safety, Faculty of Public Health, Universitas Indonesia, Depok 16424, Indonesia; 5Public Health Department, Faculty of Sports Science, Universitas Negeri Semarang, Semarang 50229, Indonesia; 6Physical Education Department, Faculty of Sports Science, Universitas Negeri Semarang, Semarang 50229, Indonesia; 7Physical Education and Sports Science Academic Group, National Institute of Education, Nanyang Technological University, Singapore 639798, Singapore

**Keywords:** screen time, playtime, COVID-19, preschoolers

## Abstract

This comparative–descriptive multi-national research examined the screen time and playtime of preschool children aged 1–6 years before and during the COVID-19 pandemic. Parents reported on the play and screen habits of preschool-aged children on the weekday and weekends using a questionnaire on the lifestyle habits of their children. Results indicated a significant difference in screen time and playtime on the weekday and weekend before the pandemic (screen time: 1.91 ± 2.40 vs. 2.16 ± 2.60 h; playtime: 3.55 ± 2.49 vs. 4.11 ± 2.58 h, both *p* < 0.05), but during the COVID-19 pandemic, only the weekday–weekend difference in screen time was significantly different (screen time: 2.87 ± 3.15 vs. 3.26 ± 3.18 h, *p* < 0.05; playtime: 3.25 ± 3.41 vs. 3.48 ± 2.41, *p* > 0.05). Before- and during-COVID-19 comparisons showed that the average daily screen time increased by 150% from 2.04 h to 3.06 h (*p* < 0.05), while the average play time decreased by 12.3% (3.83 to 3.36 h, *p* < 0.05). Based upon international guidelines for movement behaviours of young children, special attention and actions are needed to manage the excessive daily screen time and preserve the average daily playtime of Javanese preschool children. These results present useful benchmarking data for parents, teachers, and health authorities to initiate ameliorative interventions to better balance children’s screen time and playtime as Indonesia emerges from the COVID-19 pandemic to a COVID-19 endemic.

## 1. Introduction

### 1.1. Background

The COVID-19 pandemic has forced families to respond to challenges, one of which is the restrictions on public activities [1]. Having limited freedom of movement to places and spaces, people had no choice but to do mostly everything from the home. Parents play an incredibly important role in responding to the situation [2]. From pregnancy, both parents must try to reduce screen time and participate in physical activities regularly, and build an awareness of the importance of spending more time doing physical activities [3]. In reality, many parents would rather reduce their physical activities so that they can have more screen time and be involved in other activities with the family [4]. Research shows that screen time and physical activities vary by gender, with men spending more time using screen gadgets than women, and women being more physically active than men [5]. Mindful of these findings both parents must strive to be positive role models for young children as they are dependent on caring adults in helping them form health-enabling habits for optimal growth and development. A more in-depth analysis of the cited study showed that at all ages, there was a trend of increased sedentary time and reduced physical activities [6]. In support, at the beginning of the pandemic, a survey that involved 3000 parents and legal guardians with preschool-age children (average age of 3 years) showed that during the pandemic children spent less time on physical activities and had more screen time [7].

### 1.2. Research Challenges

There are strong challenges in implementing screen time and physical activity recommendations especially during the COVID-19 pandemic when schools became less accessible because of closures and children spent much time at home [8,9]. In the case of Java, Indonesia, only 39% of schools held in-person physical classes with limited student numbers between March 2020 and September 2021) during the COVID-19 pandemic, and this affected the screen and physical activity times of preschool children [10]. In addition, health governance has declared that it intends to implement restrictions on community activities; consequently, many children spend the majority of their time indoors, minimizing outdoor movement during COVID-19 interactions. The World Health Organization (WHO) has advanced 24-h movement guidelines for how preschool children should spend time for wholesome and healthy development. For children to live a healthy life, each 24 h (i.e., a day) should be filled with: (i) ≥180 min of physical activities, inclusive of ≥60 min of energetic play; (ii) ≤1 h of screen time; and (iii) 10–13 h of quality sleep [11].

For children, screen time and outdoor playtime are associated with sleep duration and sleep pattern, and therefore reducing screen time and increasing outdoor playtime will help improve their sleep [3]. Children having the recommended hours of physical activities and screen time and adhering to the recommended sleep guidelines show better control of emotions or self-regulation and also school readiness [12]. Before COVID-19, the condition of the citizenry was under control, but some of the literature concerns the effects of increasing screen time on children. As the spread of COVID-19 was very rapid, during the pandemic from February to July, Indonesia experienced its first case of COVID-19 in February, followed by thousands of additional cases. Therefore, health governance declared to implement restrictions on community activities, evaluated by area using code levels 1 through 4. Level 4 is the most restricted area for all activities, with the exception of health care. Furthermore, level 1 consists of slower COVID-19 deployment. On this day, almost all activities are restricted for people of all ages, so many individuals opt to spend their time at home. Implemented by the government, work and school from home adapt to the conditions of the virus’s spread. Consequently, this condition increases screen time for individuals, especially children. In contrast, elsewhere, in Chile, children’s physical activities declined as recreational screen time and sleep duration increased and sleep quality decreased during the early stages of the virus pandemic [7]. In support, in the South American regions, physical activities decreased by 25% while screen time doubled [13].

Other studies on screen time and playtime examined the correlation between these two variables using cross-sectional data [14,15,16] and longitudinal data [3]. Additionally, research identifying the influences of screen time and playtime on children’s social skills [17] and cognitive variables [18] is available. In the main, researchers argue that limited recreational screen time and sufficient playtime among young children are beneficial in early childhood. Many of these studies were conducted before the virus pandemic [3,4,17]. The number of studies on play activities during the pandemic is limited, with only one or two studies focusing on children’s physical activities during the pandemic [6]. The implementation of large-scale social restrictions makes data collection very challenging as access to preschools and parents became very cumbersome as many people were adjusting to a serious infectious disease.

Given these challenging circumstances for research, only a few managed to map screen time with playtime on weekdays and weekends among children before and during the COVID-19 pandemic in the Indonesian context. The purpose of the present research was to examine and elucidate the relationships between screen time and playtime on a weekday and a weekend before and during the COVID-19 pandemic in preschool children living in Java, Indonesia.

## 2. Methods

### 2.1. Research Design and Ethical Approval

The present study was a research collaboration between the universities in Java and the National Institute of Education, Nanyang Technological University in Singapore. The research was approved by the ethics committees of the Institute of Research and Community Service, Universitas Negeri Yogyakarta, Indonesia (540/UN34.21/TU/2019) and that of the Nanyang Technological University, Singapore (NTU IRB-2019-02-036). The study involved the same group of adult parent respondents living on Java island, Indonesia, as previously reported by Fauziah and her colleagues that examined the play–sleep nexus of preschool children on Java island, Indonesia [19]. Unlike the cited study, the proportion of total playtime spent that was of MVPA quality on a weekday and a weekend was examined in the present study Further, parent accompaniment during child screen time and child playtime are unique features in the present study.

### 2.2. Participants and Data Collection

The research involved 618 adult parents with preschool children aged 1–6 years who answered a content-validated and internal consistency-established online questionnaire (SMALLQ^®^) (National Institute of Education, Singapore) after giving their informed consent. Parents were recruited using a snowball sampling technique [20]. To gather information on kindergartens in the surrounding area, a team from Universitas Negeri Yogyakarta, Universitas Indonesia, and Universitas Negeri Semarang worked together to accomplish the research. Each team from a different university contacted partner kindergartens with permission to survey all adult parents/caregivers of children aged between 21–65 years in order to collect information about the daily lifestyle habits of the children before and during the COVID-19 pandemic, utilizing the snowball technique, then participants completed the surveillance instrument. No identifiers were collected in the questionnaire and all personal data remained confidential. All questions—including those on screen time and playtime of children in a 7-day recall were answered according to the instructions set out in the questionnaire. Screen time and playtime data from 618 parent-participants were used in the research analysis. The before-COVID-19 pandemic data were collected from July to October 2019 and the during-COVID-19 pandemic data were collected from February to July 2020.

### 2.3. Surveillance of Digital Media Habits in Early Childhood Questionnaire (SMALLQ^®^)

The Surveillance of Digital Media Habits in Early Childhood Questionnaire (SMALLQ^®^) is an adult-reported questionnaire on the digital and non-digital habits of children on a weekday and on a weekend with acceptable validity and internal consistency [21,22]. The details of its content validation and internal consistency are elaborated elsewhere [22]. Translation of the SMALLQ^®^ into the Indonesian language was based on the guidelines for cultural adaptation advocated by the World Health Organization [23]. The SMALLQ^®^ consisted of 25 items that enquired about the digital media and non-digital media habits of parent and child on a weekday and on a weekend and demographic information of the caregiver and child [21]. For the purpose of the present study, only those questionnaire items (10 items) on screen time and playtime were extracted for data analysis in the present study.

The internal consistency of the SMALLQ^®^ for its use in Indonesia was also established. The Cronbach alpha values for child total screen time and playtime data collected before the COVID-19 pandemic were 0.905 and 0.879, respectively. Child screen time and playtime data collected during the COVID-19 pandemic yielded Cronbach alpha values of 0.820 and 0.816, respectively. Cronbach alpha values of 0.70 and above are deemed to be valued and acceptable [24].

### 2.4. Key Variables of Interest

#### 2.4.1. Child Total Screen Time

Duration spent on screen media for entertainment (e.g., using screen media for entertainment, for media creation, or for communicating with relatives) and non-entertainment activities (e.g., using screen media for education/learning) on a weekday and weekend, respectively, were summed to provide an estimate of child total screen time over each 24 h period.

#### 2.4.2. Physical Playtime

Time spent on indoor play (e.g., dancing) and outdoor play (e.g., playing ‘hide-and-seek’ at a playground) were summed to provide an estimate of the child’s total physical playtime on a weekday and weekend day, respectively.

#### 2.4.3. MVPA Duration

Moderate-to-vigorous intensity physical activity (MVPA) is described in SMALLQ^®^ as physical activities during which breathing is faster and harder. Parents reported the percentage of playtime during which their child breathes faster and harder. The expression to derive MVPA in duration (hr) is (indoor + outdoor playtime in hr) * (% of time during which a child breathes faster and harder ÷ 100%). The duration of MVPA is calculated on a weekday and a weekend, respectively.

### 2.5. Data Analysis

Data were analyzed using SPSS version 23.0 statistical software. The analysis included data on screen habits and playtime on weekdays and weekends before (*n* = 309) and during (*n* = 309) the pandemic. Descriptive analysis (mean ± SD) was performed to describe the participant characteristics while *t*-tests (independent sample *t*-test and paired samples *t*-test) were conducted to determine whether there were significant differences in screen time and playtime on the weekday and on the weekend as well as before and during the COVID-19 pandemic. The level of statistical significance was accepted as *p* < 0.05.

## 3. Results

### 3.1. Characteristics of the Parent/Caregiver Survey Respondents

The characteristics of the parent/ caregiver who responded to the survey are outlined in Table 1. Most (83.4%) of the respondents were in their mid-twenties to thirties (between 25 and 39 years old) at the time of the survey. Mothers made up 77% of the respondents and most (70.6%) were Javanese.

### 3.2. Characteristics of Screen Time and Playtime in Preschool Children before and during the Virus Pandemic

From Table 2, it is noteworthy that weekend screen time in preschool children was 13% (0.25 h) more than on the weekday before the virus pandemic and 13.6% (0.39 h) more than on the weekday during the virus pandemic (*p* < 0.05). Weekday and weekend screen time of preschool children increased by 150% (0.96 h for weekdays and 1.10 h for weekends) during the virus pandemic compared to before the virus pandemic.

Preschool children spent modest amounts of time at playtime before and during the virus pandemic. Weekend playtime in preschool children was 15.8% (0.56 h) more than the weekday (*p* < 0.05), before the virus pandemic and 7.1% (0.23 h) more than the weekday, during the virus pandemic (*p* > 0.05). Weekday and weekend playtime of preschool children decreased by 8.5% and 15.3%, respectively, during the virus pandemic compared to before the virus pandemic

Table 3 shows the proportion of screen time for entertainment and for non-entertainment, on the weekday and weekend, before and during the virus pandemic. Screen time for non-entertainment and for entertainment increased significantly on the weekday and on the weekend before and during the virus pandemic (*p* < 0.05). Before and during the virus pandemic, entertainment-based screen time was 2.8 to 4.8 times that of non-entertainment-based screen time on the weekday and on the weekend.

Table 4 presents information about the quality of play (i.e., % of time that is MVPA) and the proportion of time parents engaged with their children during playtime. Juxtaposing data presented in Table 2 and Table 4, time spent in MVPA before-COVID-19 pandemic was 1.88 h on the weekday and 2.54 h on the weekend, and during the COVID-19 pandemic was 1.79 h on the weekday and 2.19 h on the weekend.

Parent accompaniment during child screen time and child playtime were shown in Table 3 and Table 4. On weekends, parent accompaniment during child screen time during COVID-19 was significantly higher than parent accompaniment before COVID-19 (62.6% vs. 56.6%; *p* < 0.05). Parent accompaniment during child screen time was higher on a weekend than on a weekday (before COVID-19: 50.3% vs. 56.6%; during COVID-19: 52.6% vs. 62.6%). Similarly, during child playtime parent accompaniment was higher on a weekend than on a weekday (before COVID-19: 63.5% vs. 44.9%; during COVID-19: 62.4 vs. 49.1%).

## 4. Discussion

The primary purpose of the present research was to examine the relationships between screen time and playtime on a weekday and on a weekend, before and during the COVID-19 pandemic in Indonesian preschool children living in Java. For children in early childhood, the American Academy of Pediatrics (AAP) stipulates no screen time at all for children until 18 to 24 months, except for video chatting, and adds that children aged 2 to 5 years should have an hour or less of screen time per day [25]. In the context of physical activity (play), sedentary behaviour (screen time) and sleep, the World Health Organization publicized that over a 24 h period, preschool children under 5 years should spend (1) ≥180 min doing physical activities including ≥60 min of energetic play; (2) ≤1 h of screen time; and (3) 10–13 h of quality sleep [11].

### 4.1. Screen Time in Preschool Children before and during the Virus Pandemic

From Table 2 and Table 3, the screen time of preschool children living on Java island already exceeded the WHO 2019 guideline for sedentary screen time of not more than 1 h per 24 h cycle by 0.91 h on a weekday and 1.16 h on the weekend before the virus pandemic. This finding is exacerbated during the virus pandemic where screen time exceeded the WHO guideline by a staggering 1.87 h and 2.26 h, on a weekday and on a weekend, respectively. The present data also showed that entertainment-based screen time was 2.8 to 4.8 times that of non-entertainment-based screen time, with this difference more apparent over the weekend.

Data elsewhere support our present findings of greater screen time in young children during the virus pandemic. For instance, in a national sample of toddlers and preschool children, Jáuregui and colleagues reported higher screen time in preschool children living in Mexico, during the virus pandemic lockdown, especially in those (i) with ready access to gadgets, (ii) came from economically disadvantaged backgrounds and (iii) those who were without playmates during the virus pandemic. In the cited study, physical activity decreased by 25%, screen time doubled, and sleep quality declined by 17% [13].

The importance of setting limits for daily screen time, especially for entertainment in preschool children is a refrain echoed by many educationalists as excessive sedentary screen time portends greater fat storage in the body and hence contributes to juvenile obesity [26,27]. Excessive screen time in early childhood, and in combination with insufficient sleep and physical activity, has also been associated with poorer executive function [28], socio-emotional function [29], lower cognitive development [30], poorer motor function [18], and expressive language delays [31]. Additionally, it appears that excessive screen exposure in 20324 preschool children living in Shanghai, China was associated with poor psychosocial well being via a number of mediators, mostly by reducing parent–child interaction [32]. Some meta-analytical research demonstrates negative associations between television viewing and the quality of parent–child interactions [33].

Evidence for the detrimental effects of excessive screen time in preschool children is emerging. For instance, Xie and his colleagues reported preschool children exposed to more than 60 min of screen time per day faced higher risks of behavioural problems such as inattention, temper control, and presenting symptoms of attention deficit disorder. Further, others suggest that excessive screen time in early childhood increases the likelihood of less creative play, poorer vocabulary, and delayed attainment of developmental milestones [34].

Several explanations are offered for the increased prevalence of excessive screen use in preschool children. For instance, because of the COVID-19 pandemic and associated movement and travel restrictions, families were ‘compelled’ to go online a lot more. Additionally, with the need for education to continue, schools were encouraged to conduct online classes instead of in-person ones [35].

Parent attitudes towards their own screen use and that of their children are mediators of overall familial screen time. For instance, Chia and his colleagues reported parents’ own screen use predicted child screen use and children ranked in the top quartile for daily screen time tended to come from households where parents had more ‘liberal’ attitudes towards screen time and tended not set time limits for child use of screens [22]. The explanation for parent attitudes towards screen use is instructive.

These days, many parents have built a habit or routine of prolonged screen time at home to help their children learn [35]. However, while a majority of parents reported that they held the intentional attitude that child screen time was useful for education and for learning, in reality, and in practice, they also reported that the child’s screen time was mainly used for entertainment [36]. This finding is also true for the present study where screen-based entertainment for children was a daily substantive indulgence for preschool children before and during the virus pandemic.

Others recognize the utility of screen time for families seeking relief and restoration from stressors during the pandemic—that children are kept occupied and entertained while eating [37] or when parents/caregivers attended to errands or household chores [38]. In this regard, supporting parents/caregivers in selecting high-quality educational apps that are evidence-based to support learning should be considered [39].

There is support for the key finding of increased child screen use in the present study. For instance, Bergmann and his colleagues investigated children’s screen time (*n*= 2209, age 0.7–3 years; caregiver-reported), sampled from 15 laboratories across 12 countries during the COVID-19 pandemic. They explained that toddlers with no online schooling requirements, experienced increased screen exposure during the lockdown periods of the COVID-19 pandemic, with longer screen exposure going hand in hand with the severity of the lockdown. They reported further that child screen time during the virus lockdown was negatively associated with the socio-economic status of the caregivers and was positively associated with child age, caregiver screen time, and attitudes towards children’s screen time [40].

The cited results are further affirmed by similar research accomplished in Singapore, albeit just before the virus pandemic struck, where high screen time among preschool children was already prevalent and where only 9.6% of children met all 3 WHO guidelines on physical activity, sedentary behaviour, and sleep [41]. Therefore it does seem that the high screen time in early childhood already present before the virus pandemic is exacerbated even more by the impact of the COVID-19 pandemic.

Another plausible explanation for the high screen time of preschool children before the virus pandemic and even higher screen time during the virus pandemic is high caregiver or parental stress. In a study that involved parents of older children aged 6–12 years, researchers explained parents experienced moderate to high stress levels, especially during school closures and that higher parental stress was associated with increased hours of screen use by children while increased parental involvement was associated with a smaller increase in child screen time during the virus pandemic [42]. This explanation is also supported by Okely and his colleagues who researched the global impact of the virus pandemic across 14 countries [43]. They surmised that screen time among 3–5 year old children increased during the virus pandemic for several reasons—parents working from home and using screen time to keep children busy while they worked, children going online for school activities, and children playing less outdoors and also using screen time to connect with other family members who were restricted from travelling home.

### 4.2. Playtime in Preschool Children before and during the Virus Pandemic

Data presented in Table 2 suggest that Javanese preschool children spent modest amounts of time at play before and during the pandemic. These playtime durations were greater than the 180 min of daily play recommended by WHO. Nonetheless, there was a slight dip of 1.7% for weekday and 8.4% for weekend playtime during the virus pandemic compared to before the virus pandemic. In the context of Javanese preschool children, these playtime habits need to be supported and sustained since high-quality playtime, especially those that involve hands-on parent or caregiver–child interactions such as ‘serve and return’ activities confer many health benefits—positive child conduct when older, emotion regulation, understanding, and creativity [44].

‘Serve and return’ activities are those where children initiate interest in an activity and parents respond appropriately (serve and return parenting) is enriching because it supports children’s learning and social skills development [45]. Child hands-on play indoors (e.g., craftwork, drawing, colouring, jigsaw puzzle assembling, and role-play activities) or outdoors (playground activities such as climbing, tag games, running, throw and catch, and energetic play) especially when accompanied by parent or caregiver are opportunities for high-quality interactions that promote nurturing and caring bonds between parent and child—creating meaningful memories that could linger beyond the early childhood years. In the present study, parents spent between 49% and 64% of playtime with their children. These sessions allow parents to forge meaningful memories with their children and quality parent–child interactions can certainly provide good foundations for good familial relationships.

Studies elsewhere show mixed results on the impact of the virus pandemic on outdoor play in children. For example, some studies emanating from Germany [46] and the Netherlands [47] showed an increase in outdoor play in children during the virus pandemic lockdown while children in Canada [48] reduced outdoor play during the lockdown period. These mixed results could plausibly be explained by country differences in policies and/or typical housing and the neighbourhood environment (e.g., access to safe playgrounds and public parks). Research in others countries that examine the impact of the virus pandemic on screen time and playtime in preschool children presents several key points for consideration. In a longitudinal study across 14 countries that sampled 948 preschool children aged between 3 and 5 years old, which examined the changes in movement behaviours (including screen time and physical activity), Okely and his colleagues reported small declines in the playtime of preschool children [43], in agreement with the results of the present study. Additionally, these researchers deduced that having access to spaces for play such as having a compound within the house or having access to safe neighbourhood playgrounds and supporting parents’ mental health could help sustain children’s playtime through the virus pandemic.

### 4.3. Strengths and Limitations

The present study examined two equal-sized samples of parents/caregivers in comparing the screen time and playtime of children aged between 1 and 6 years living on Java island before and during the virus pandemic using an appropriate validated and reliability-tested questionnaire. The research provided context- and situation-specific data that could be used either for further research or serve as rich data to plan for joint ameliorative programmes between parents and the schools to address the demerits of the excessive use of screen time and also to better preserve high-quality interactive outdoor play among parents and preschool children. Several limitations of the present research are noteworthy—the results obtained were from parents’ self-reports and these were subjective responses that could be threatened by recall and social desirability biases. These biases were mitigated and minimized, respectively, by having the parent recall of child behaviours limited to the last 7 days and keeping the survey anonymous. Research suggests that parent’s self-reports of children’s PA (duration and intensity) may be different from those using objective tools such as motion sensors (accelerometers) as child-proxy reports by adults tend to over-estimate physical activity and under-estimate sedentary time so interpretation of data from different methods used to estimate activity and non or low activity should be made with caution [49]. Indeed going forward, the use of a combination of both objective and subjective methods could provide a more holistic picture of the activity behaviours of young children.

Another limitation of the research is that as Indonesia is a very large country, the results were focused on parents and children living on Java island and cannot be generalized beyond the sample. Future studies should focus on whether these changes were altered as the restrictions from the pandemic were relieved.

## 5. Conclusions

In summary, taken as a whole, the results of the present study on preschool children living in Java island, Indonesia showed the virus pandemic has altered people’s lives in ways that were not previously anticipated—children’s engagement with screens increased significantly while children’s playtime showed a modest dip, compared to before the virus pandemic. In view of the present findings, the authors recommend that parent education efforts coupled with parents working in concert with teachers to forge alliances where culture- and context-specific strategies, ideas, and practical tips can be exchanged and tried out in ameliorating lifestyle behaviours—namely screen time and playtime among preschool children living in Java island. The benefits and demerits of media and social screen time should be further explored in relation to the mental health outcomes of parents and preschool children during the ongoing COVID-19 pandemic. Further, to promote high-quality parent–child interactions and positive developmental outcomes during the pandemic and beyond, childcare needs and parent well-being should be supported.

## Figures and Tables

**Table 1 ijerph-20-01659-t001:** Demographical characteristics of the adults who responded to the survey.

Parameter	Pre-COVID Groups	During-COVID Groups	All Groups	
			Total	%
Child Age (Years)				
6	48	50	98	15.9
5	96	68	164	26.5
4	85	65	150	24.3
3	49	51	100	16.2
2	25	63	88	14.2
1	6	12	18	2.8
Parent/Guardian (Relationship)				
Mother	271	205	476	77.0
Father	27	66	93	15.1
Grandmother	3	17	20	3.2
Grandfather	2	3	5	0.8
Legal Guardian	6	18	24	3.9
Parent/Guardian Age (Year)				
16–19	0	2	2	0.3
20–24	19	14	33	5.3
25–29	71	72	143	23.1
30–34	120	127	247	39.9
35–39	67	59	126	20.4
40–44	26	22	48	7.8
45–49	2	9	11	1.8
50–54	2	2	4	0.7
55–59	1	1	2	0.3
60–64	1	1	2	0.3
Parent/Guardian (Race/Ethnicity)				
Jawa	295	141	436	70.6
Sunda	6	73	79	12.8
Betawi	0	26	26	4.2
Batak	1	14	15	2.4
Others	7	55	62	10.0

**Table 2 ijerph-20-01659-t002:** Screen time and playtime of preschool children on a weekday and a weekend before and during the COVID-19 pandemic.

Behaviour	Period	Mean ± SD		t	Sig.
		Weekday (hours)	Weekend (hours)		
Screen Time	Pre-COVID-19	1.91 ± 2.40	2.16 ± 2.60	−2.975	0.003
	During COVID-19	2.87 ± 3.15	3.26 ± 3.18	−2.730	0.007
Playtime	Pre-COVID-19	3.55 ± 2.49	4.11 ± 2.58	−5.891	<0.001
	During COVID-19	3.25 ± 3.41	3.48 ± 2.41	−1.707	0.089

Note: Statistical significance is set at *p* = 0.05. Paired samples *t*-tests were run to examine differences between weekdays and weekends.

**Table 3 ijerph-20-01659-t003:** Non-entertainment and entertainment screen time of preschool children and parent accompaniment in child screen use before and during the COVID-19 pandemic.

Characteristics of Child Screen Time	Group	Mean hrs	95% CI			Sig.
			Lower	Upper	*t*	
Non-entertainment screen time on a weekday (hrs)	pre-COVID-19	0.42 ± 0.77	−0.510	−1.636	−3.823	0.001
	during COVID-19	0.76 ± 1.34				
Entertainment screen time on a weekday (hrs)	pre-COVID-19	1.49 ± 1.92	−0.961	−0.286	−3.628	0.001
	during COVID-19	2.12 ± 2.34				
Non-entertainment screen time on a weekend (hrs)	pre-COVID-19	0.40 ± 0.74	−0.297	−0.028	−2.375	0.018
	during COVID-19	0.56 ± 0.95				
Entertainment screen time on a weekend (hrs)	pre-COVID-19	1.76 ± 2.20	−1.34	−5.26	−4.504	0.001
	during COVID-19	2.70 ± 2.90				
% of time parent engaged in child screen use (weekday)	pre-COVID-19	50.3 ± 32.9	−7.479	2.923	−0.860	0.390
	during COVID-19	52.6 ± 32.7				
% of time parent engaged in child screen use (weekend)	pre-COVID-19	56.6 ± 34.7	−11.307	−0.641	−2.200	0.028
	during COVID-19	62.6 ± 32.7				

Note. Time spent using screen media for entertainment, for media creation, for communicating with relatives, and other activities was summed up as entertainment screen time. Time spent using screen media for education/learning was considered non-entertainment screen time. Independent sample *t*-tests were run to compare between groups (pre-COVID-19 and during COVID-19).

**Table 4 ijerph-20-01659-t004:** Proportion of child playtime that is of moderate-to-vigorous intensity and the proportion of playtime parent was engaged with children before and during the COVID-19 pandemic.

Characteristics of Child Playtime	Group	Mean	95% CI			Sig.
			Lower	Upper	*t*	
% of playtime is of MVPA (weekday)	pre-COVID-19	53.0 ± 27.1	−6.299	2.318	−0.907	0.365
	during COVID-19	55.0 ± 27.3				
% of playtime is of MVPA (weekend)	pre-COVID-19	61.8 ± 28.4	−5.322	3.409	−0.430	0.667
	during COVID-19	62.8 ± 26.8				
% of time parent engaged with child during play (weekday)	pre-COVID-19	44.9 ± 26.4	−8.551	0.175	−1.885	0.060
	during COVID-19	49.1 ± 28.8				
% of time parent engaged with child during play (weekend)	pre-COVID-19	63.5 ± 28.5	−3.418	5.668	0.486	0.627
	during COVID-19	62.4 ± 29.0				

Note. Moderate-to-vigorous intensity physical activity (MVPA) is described in SMALLQ^®^ as physical activity which makes a child breathe faster and harder.

## Data Availability

Data can be made available upon written request to the corresponding authors.

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
