# Peer review of "Time to Play in Javanese Preschool Children—An Examination of Screen Time and Playtime before and during the COVID-19 Pandemic"

_ijerph, 2023, doi:10.3390/ijerph20031659_

Round 1
Reviewer 1 Report
Time to Play in Javanese Preschool Children – An Examination 2 of Screen Time and Playtime Before and During The Covid-19 3 Pandemic
Thank you for an interesting and well written manuscript. I have some comments below, mostly about clarifications in the text.
Line 40-42: The sentence that a mother must reduce screentime and increase physical activity seems a bit old-fashioned – why should not fathers do the same? I understand why you have put emphasis on the mothers role, since the study you refer to is based on mothers experience. Although, the study you refer to looks into sceen time and physical activity and impact on sleep pattern and behaviour. And the next sencente about men having more sceen-time and less physical activity time does not match your statement about mothers. Could you modify or remove this sentence, for me it gives an old-fashioned feeling that does not match the values in modern society.
Line 104: It says that children 2-5 years old were included, but the abstract says children 1-6 years.
Line 112: The covid-data was collected from feb to July 2020, could you possibly add a comment about what the restrictions were like in Java at that time point. The restrictions were a bit different over the world, and the pandemic was spread a bit differently timewise. Just for readers to get a clearer picture of the relevance of the time frame for the study.
Line 113: for this section, could you add information about the number of items in the SMALL questionnaire, and the number of items used in the study.
Table 2: Can you specify in the table, that the time reported is in hours, for activity, I cannot find it in the table.
Line 330-341: This section seems a bit out of topic for me, could it be shortened some? I believe the relation to the results of your study could be stated without presenting the (for reader in this manuscript) new term “serve and return”, and the explanation of it.
Line 374-380: I would like to see an extended discussion about the pros and cons of parent reported PA and MVPA, in comparison to objective measures, as accelerometers. Are there studies comparing the two methods? How reliable are the results of the PA and MVPA that is exceeding the WHO recommendations by far? Also, to what extent did the children included go to kindergarten or similar facilities? Can the estimated time in PA and MVPA be affected of the fact that parents and children are not together for a significant amount of time during the weekdays? I would like to se a discussion about that.
Line 381-394: I believe that the sentence on line 383-84 should be modified some, since it can be interpreted as that the change in habits during the pandemic are permanent, although your study shows that changes were found from before and during the pandemic. Future studies should focus on whether these changes were altered as the restrictions from the pandemic were relieved.
References:
Just a few references are older, can the following ones be updated by later research?
#2 and #28
Author Response
-

Reviewer 2 Report
The article on " Time to Play in Javanese Preschool Children – An Examination of Screen Time and Playtime Before and During the Covid-19 Pandemic" describes research important for behavioral sciences and practice in young childhood. But the topic is lack of novelty. Also, I have few major concerns about study subjects, data analysis and interpretation.
1. Please check the sample size. The number of study subjects were described inconsistently in methods section, Table 1, and discussion section. Please see line 104, Table 1 and line 368. Furthermore, the Data Collection describes only adult parents with preschool children aged 2-5 years were recruited in this study. But why few numbers of children aged ≤1 year were described in Table 1? If this study only included children aged 2-5 years, please exclude these study participants.
2. This study majorly compared the differences of screen time and physical activity duration between pre- and during COVID, but independent study subjects were recruited in this study. The demographic factors in Table 1 should be checked if any differences between pre- and during COVID group and adjusted in further analysis to obtain robust results.
3. Please discuss these findings according to Java Island’s situation due to different policy restrictions and the number of COVID-19 infections among countries that directly affected behavior.
Minor:
Introductions
1. Information on background during COVID period in Java Island, Indonesia is unclear.
Methods
1. How were study participants recruited in this survey? Can this study population represent whole Java Island population who aged 2-5 years?
Results
1. Are there any other demographic factors available in this study, such as gender, number of siblings, social economic status? Based on prior findings, these factors may play important roles to associate with child’s behavior.
2. Several information are unclear, such as did these study subjects go to daycare before COVID and switch to virtual course during COVID or still go to daycare in-person because there were still 39% of schools held in-person classes with limited student numbers between March 2020 and September 2021.
3. Tables need to reorganize, such as name of parameter in Table 1, units in Table 2.
4. The meanings of 95% CI and p value are the same. Please choose one to display in Tables.
Discussion
1. Overall editing required. Detailed results description and Table X should not be showed in the discussion. Discuss the findings instead of repeating them again.
2. Lines 213-222 should be removed. Same information has been indicated in study purpose and methods. Please describe the main findings of this study in the beginnings of discussion section.
3. Lines 224-230, screen time before COVID pandemic was already above the WHO suggested screen time in this population. Any reason can explain this finding?
4. Many studies have investigated the impact of COVID-19 on children’s PA and ST. Compared to previous studies, what is the importance and strength of the current study?
Author Response
-

Round 2
Reviewer 2 Report
Dear Authors,
I have only few minor points. Please find specific comments below:
1. Please check the abbreviation has been shown at the first time, such as the World Health Organization.
2. I suggest to move the paragraph (lines 125-135) from Participants and data collection to Introduction section.
3. Please clarify the sentence “Each team from a different institution visited partner kindergartens with permission to survey all adult parent/caregiver of children aged between 12–65 years in order to collect information about the daily lifestyle habits of the children before and during COVID-19 pandemic.” Did study staff collection data in person? If yes, how to collection data from parent/caregiver of children if kindergartens closed during COVID pandemic?
4. Please check spelling.
Author Response
I have included our confirmation and reply comments regarding how to enhance each section of the work. We appreciate all improvements you make to the article that enhance its readability. Warm regards.
Rizki Mulyawan.
